# Building Trusted Federated Learning on Blockchain

**Yustus Eko Oktian** , **Brian Stanley and Sang-Gon Lee ***

College of Software Convergence, Dongseo University, 47 Jurye-ro, Sasang-gu, Busan 47011, Korea;
d0185099@kowon.dongseo.ac.kr (Y.E.O.); briansta1601@gmail.com (B.S.)
* Correspondence: nok60@dongseo.ac.kr

**Abstract:** Federated learning enables multiple users to collaboratively train a global model using the users' private data on users' local machines. This way, users are not required to share their training data with other parties, maintaining user privacy; however, the vanilla federated learning proposal is mainly assumed to be run in a trusted environment, while the actual implementation of federated learning is expected to be performed in untrusted domains. This paper aims to use blockchain as a trusted federated learning platform to realize the missing "running on untrusted domain" requirement. First, we investigate vanilla federate learning issues such as client's low motivation, client dropouts, model poisoning, model stealing, and unauthorized access. From those issues, we design building block solutions such as incentive mechanism, reputation system, peer-reviewed model, commitment hash, and model encryption. We then construct the full-fledged blockchain-based federated learning protocol, including client registration, training, aggregation, and reward distribution. Our evaluations show that the proposed solutions made federated learning more reliable. Moreover, the proposed system can motivate participants to be honest and perform best-effort training to obtain higher rewards while punishing malicious behaviors. Hence, running federated learning in an untrusted environment becomes possible.

**Keywords:** federated learning; artificial intelligence; blockchain; smart contract

## 1. Introduction

Many companies or organizations have recently utilized Machine Learning (ML) to gain knowledge from their data. These data are mainly obtained from users when they use companies or organizations' products in their daily lives. The more data gathered from the users, the more accurate the company analytics may become, further driving the data collection practice; however, this data collection often faces public scrutiny from the user's side, as data privacy has gained public awareness lately (e.g., through GDPR law [1]). Thus, a privacy-preserving ML scheme must be built to comply with the user's data privacy requirement.

Federated Learning (FL) [2] allows ML models to be trained in users' local devices instead of companies' centralized servers; hence, avoiding the user data gathering in the first place; however, the vanilla FL is assumed to be run in a trusted environment, where the trainers are always honest. Meanwhile, the real applications of FL are in an untrusted domain, in which trainers can become malicious. This mismatch highlights the necessity of a secure platform to run FL, where multiple conflicting participants can train models honestly and fairly.

On the other hand, blockchain technology has gained traction lately due to the popularity of Bitcoin [3]. The premise of blockchain is to allow users to store data or process data (e.g., through smart contract [4]) securely in a distributed manner without third-party intervention. If we look closely, the "distributed storage and computing" of blockchain are what we need for "decentralized training" in FL; therefore, many researchers have adopted blockchain to their FL system to secure the collaborative training efforts made by users [5].

Driven by the same background, we propose a blockchain-based FL system in this paper. Our motivation to create yet another blockchain FL system is that most previous papers only use blockchain as a platform to store and audit the trained models. Meanwhile, the FL system is a complex collaboration system involving many other issues, such as how to motivate clients to perform training, how to ensure the quality of the trained model, and how to guarantee the security and fairness of the FL system. So far, only few studies have addressed those issues [6–8]. Our paper exists to (i) solve those unresolved issues and (ii) provide alternative solutions from those mentioned studies.

In summary, our contributions are as follows.

- We analyze several issues of vanilla FL such as low motivation to perform local training, client dropouts, model poisoning, model stealing, and unauthorized access.
- Based on the previously mentioned problems, we create initial building blocks for our proposal, including an incentive mechanism, peer-to-peer reviewed trained model, model encryption, stage timeout, deposit, and reputation system.
- We design a full-fledged blockchain-based FL protocol to run fair, secure, and trusted FL tasks.
- We provide a proof-of-concept implementation of our protocol and analyze the results.

The rest of this paper is organized as follows. We first explain our proposal's problem statement and building blocks in Section 2. We then explain the inner workings of our proposal in Section 3 and evaluate our proposed method in Section 4. Literature reviews on the state-of-the-art blockchain-based FL protocols are presented in Section 5. Finally, we conclude in Section 6.

## 2. Preliminaries

### 2.1. Problem Statement

The vanilla FL [2] still relies heavily on a centralized and trusted environment. Because of that, we find several general problems (GP) as follows.

**GP1** The model owner cannot recruit enough workers to train their model due to a lack of incentives for workers.

The success of FL training depends heavily on the workers' willingness to perform the FL tasks. The model may become less accurate if it is trained with only a few numbers of data. On the other hand, having more workers may result in more data available for training and increase the data diversity, assuming that each worker trains the model with a unique dataset. Unfortunately, from the workers' point of view, performing local training means wasting their resources. The workers will most likely not perform the FL tasks in the vanilla FL because it has no incentive mechanism to attract workers' participation.

**GP2** The workers may perform malicious local training that may corrupt the global model.

Because all local models will be aggregated into a global model, corrupt or invalid local models may disrupt the global model. The vanilla FL does not have any protection against malicious workers. The lazy workers, which perform training with low effort, may generate a low accuracy of local models that may decrease the overall global model accuracy [9]. Malicious workers may perform the local model training with adversarial examples to make the global model misclassify [10]. Finally, workers who joined the FL tasks may suddenly drop out and discontinue the training process. When many workers stop the local training simultaneously, it may drastically affect the quality of the updated global model [9].

**GP3** Malicious actors may gain access on the updated local or global models.

FL encourages model sharing among entities, where models will be passed on from one entity to another. An unauthorized party may steal the global or local models by eavesdropping on the communication channels. When the attacker steals the local model, they can slightly modify the models (e.g., train the stolen model with their dataset for a few epochs more), then claim the model as theirs and submit it to the system. Furthermore,

they can even resubmit the stolen model without any modification, which increases their advantages. If the attacker can obtain the global model, they hit the jackpot and obtain the most reward without training. Since vanilla FL focuses on the training process, they do not protect the communication channels; therefore, the vanilla FL is susceptible to those mentioned attacks.

**GP4** The FL actors may crash and jeopardize the whole training process.

Similar to any collaboration system, we cannot guarantee that all participants behave as intended at all times in FL. The local workers may crash and not submit the training result for a given round on time. The aggregator server may be down and fail to generate a global model for a given round. Finally, the model owner may crash and cannot send the training rewards to workers. The vanilla FL will likely fail when such circumstances happen because it does not consider its robustness and fault tolerance.

### 2.2. Design Considerations

We came up with several design decisions to solve those mentioned general problems.

### 2.2.1. Incentive Mechanism

The incentive mechanism is one of the features to solve **GP1**. With enough rewards, a particular FL task should appeal to workers and ease the model owner in finding worker candidates. An important detail regarding designing an incentive mechanism is that its design should be fair and reliable to drive its trustability. In particular, the system must reward workers according to their contributions to the global model. For example, if we reward workers using flat rewards, hard-working workers may feel that they compensate others. Most workers then become reluctant to train a model using their maximum capabilities as they will obtain the same reward as the low-effort ones.

### 2.2.2. Peer-Reviewed Local Model

All submitted local models must be reviewed to assess their quality and solve **GP2**. For this purpose, we can employ third-party reviewers to perform model validations.

After the local model training completes, the workers must disclose their training results to the reviewers. The reviewers must evaluate whether the submitted local model generates lower accuracy based on their test dataset. A low accuracy model may indicate a poor training effort or an imperfect training dataset. The reviewers must also try to aggregate the evaluated local model with the current global model to see whether the aggregation improves or reduces the model quality. A quality degradation may point to data poisoning was taking place during the training. After completing the evaluation, the reviewers must submit their evaluation score over trained local models to the system. The system then determines each worker's contributions based on the submitted training and evaluation scores.

During the evaluation, malicious reviewers (e.g., colluding with or paid by the worker) may send a fake evaluation score that does not represent the evaluated models. Hence, the system must guarantee that the fake scores will not influence the contribution scores. Moreover, a lazy reviewer can do nothing but wait for other reviewers to reveal and submit their evaluation scores to the system. After that, the reviewers copy (or modify slightly) the evaluation score and submit the stolen score as theirs; therefore, the system must also prevent such stealing possibilities from happening.

### 2.2.3. Model Encryption

The encryption over the distributed models is required to solve **GP3**. FL actors can perform encryption on the application level (e.g., encrypting the model parameters directly) or build a secure channel using the TLS protocol. With encryption, outside entities will not understand the global or local models being exchanged in the system; therefore, we can minimize the potential model leakage from outside parties.

#### 2.2.4. Stage Timeout

We usually employ stages to manage the training state in synchronous FL. In particular, the system creates a timeout in which participants must complete their work before the given deadline. Furthermore, the system will not count late contributions, thus, ignoring failed or unresponsive FL actors and eliminating **GP4**.

#### 2.2.5. Deposit Mechanism

We can impose a deposit mechanism to economically punish malicious actors and help solve **GP2** and **GP4**. When the system detects malicious activity, it penalizes all culprit actors by depleting their deposits. This mechanism is arguably a simple yet effective approach; however, a static flat deposit system may not appeal to participants.

A dynamic deposit mechanism can provide perks to attract trustable actors to join our system. For example, the more trustable the actor is, the lesser deposit he or she needs to join the FL tasks. With this lower deposit, actors can join more FL tasks, gaining more profits from those tasks. This method is possible because solving machine learning problems is similar to solving a black-box system. We do not know whether our data can produce optimal accuracy results without a trial and error approach; thus, having the benefits of joining more FL tasks with smaller deposits will be more desirable. This approach should also encourage actors to always behave honestly to maintain their lower deposit benefits. Hence, it indirectly solves **GP1**.

#### 2.2.6. Reputation System

Without a reputation system, the system will always treat all participants equally. A veteran player is indistinguishable from a new player because there is no way to know the history of player activities and assess their credibility. While this "join and forget" nature may be helpful to preserve the actors' privacy, attackers become more easily able to perform Sybil attacks in this environment; therefore, the system must determine which actors are considered veteran players and propose more benefits to them when joining our FL tasks.

However, veteran players are not always trustable. In particular, they may intentionally or unintentionally (e.g., the account is hacked by hackers) become malicious. Hence, the reputation system must also track actors' activity history in the environment and determine whether they are currently in good or evil behavior. This history of activities must become one factor in determining a given actor's credibility.

### 3. Proposed Protocol

Our proposed framework is shown in Figure 1 and Table 1 presents the description of important notations and variables in this paper.

**Table 1.** List of notations used in this paper.

| Notation | Description |
|---|---|
| $\mathcal{S}, \mathcal{O}, i, j$ | Smart contract, Model owner, Client, Reviewer |
| $l, l^{min}, C, C^{min}$ | Level, Minimum level, Credit score, Minimum credit score |
| $\tau^{id}, \tau^{desc}, \tau^{target}$ | Task id, Task description, Task target |
| $M^{(0)}, M'$ | Initial global model, Updated local model |
| $d^{train}, d^{test}$ | Train dataset, Test dataset |
| $T^{score}, T^{eval}$ | Training score, Evaluation score |
| $x, d$ | Accepted training score, distance of evaluation score towards $x$ |
| $D, \beta$ | Total deposit, Base deposit value |
| $R, R^{train}, R^{eval}$ | Total reward, Reward pool for workers, Reward pool for reviewers |
| $t^{now}, \eta$ | Current epoch timestamp, Random string nonce |
| $\Upsilon^{commit}, \Upsilon^{ipfs}$ | Commitment hash, IPFS hash generated from $IPFS(\cdot)$ method |
| $H(J)$ | Hash payload $J$ using KECCAK-256 hash function |

**Table 1.** *Cont.*

| Notation | Description |
|---|---|
| $E_k(J)$ | A symmetric encryption of payload $J$ with secret key $k$ |
| $D_k(L)$ | A symmetric decryption of payload $L$ with secret key $k$ |
| $PKE_{PK}(J)$ | A public-key encryption of payload $J$ using public key $PK$ |
| $PKD_{SK}(L)$ | A public-key decryption of payload $L$ with private key $SK$ |
| $SIGN_{SK}(J)$ | Generate public-key signature of payload $K$ using private key $SK$ |
| $VER_{PK}(L, S)$ | Verify whether the digital signature $S$ of payload $L$ using public key $PK$ |
| $X \parallel Y$ | A concatenation of $X$ with $Y$ |
| $\bigcup_{n=1}^{N} X_n$ | A concatenation of data $X$ for all index $n$. |
| $F_{A \to B}(m)$ | Transfer $m$ amount of funds from entity $A$ to $B$ |
| $IPFS(J)$ | Store payload $J$ to the IPFS network |

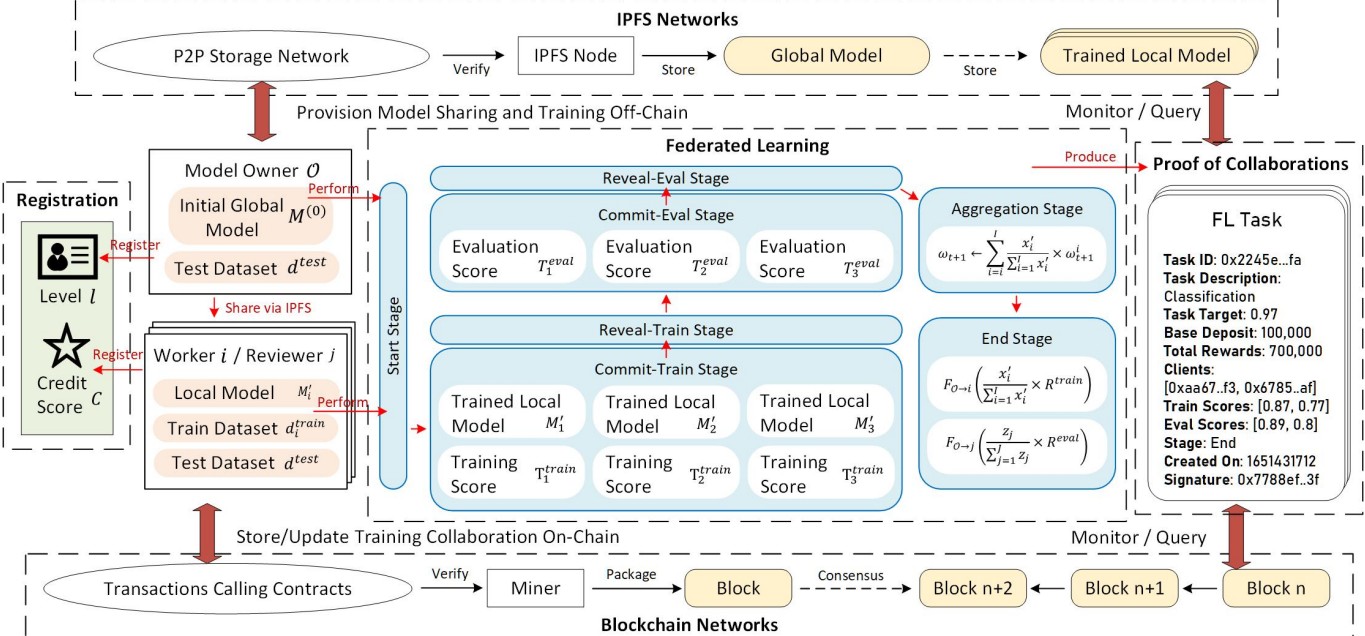

**Figure 1.** The proposed blockchain-based FL collaboration architecture, which includes seven stages. IPFS network is used to share the models between model owners, clients, and reviewers. All FL task interaction will be recorded in the blockchain and serve as proof of collaborations.

*Objectives*: Model owners want to train their global model but do not have the necessary data and hardware to perform training; therefore, they outsource the training process to multiple clients using Federated Learning (FL). Clients are willing to use their private data and resources to train the model for given incentives. The training scores and trained models from clients will be reviewed, and rewards will be distributed according to the evaluation scores. This way, model owners can be guaranteed to receive high-accuracy models while clients are compensated based on their performances. Hence, a win-win solution for model owners and clients.

### 3.1. Accountability Metric

We design our FL environment to be public such that anyone can join the FL tasks; therefore, we must guarantee that only credible clients join and perform the task to produce accurate global models. For this reason, we use two metrics to assess clients' trustability: leveling and credit score mechanism.

#### 3.1.1. Leveling System

The leveling system is used to judge the clients' experience of performing the FL tasks, which is summarized in Algorithm 1. The admin first sets $\gamma$ value, a multiplier to determine

how many experiences are required to reach the next level $e^{next}$. New client will be given zero experience ($e = 0$), which puts them into level one ($l = 1$) by default. Every time clients successfully perform a task (e.g., training or evaluating), the GAINEXPERIENCE($\cdot$) method will increase their experiences. The method also increments clients' level if the required experience to level up is reached. Higher-level clients mean that they are veteran players and are most likely to be trustable.

---

**Algorithm 1** Leveling system for client $i$

---

1: **on** startup:
2:     admin initiates $\gamma = 5$              $\triangleright$ experience growth multiplier
3:     for each new client, admin initiates:
4:         $l = 1$ and $e = 0$            $\triangleright$ default level and experience
5:         $e^{next} = l * \gamma$            $\triangleright$ experience required to level up

6: **procedure** GAINEXPERIENCE($i$)
7:     get the current experience $e$ for client $i$
8:     $e \leftarrow e + 1$
9:     **if** $e = e^{next}$ **then**              $\triangleright$ level up is possible
10:         $l \leftarrow l + 1$
11:         $e^{next} = l * \gamma$
12:         $e = 0$
13: **function** CALCULATELEVEL($i$)
14:     **return** get the current level $l$ for client $i$

---

### 3.1.2. Credit Score System

Unfortunately, high-level players are not always trustworthy because they can still act maliciously, intentionally, or not (e.g., the account is hacked by attackers). Because of that, we propose a credit score mechanism as our second trustability factor, which will monitor the history of clients' honest/malicious activities throughout their lifetime. We summarize the logic in Algorithm 2.

On startup, the admin first needs to set three parameters: $C^{evt}$, $P$, and $t^{expired}$. The $C^{evt}$ is a set of credit events containing credit values given when the client performs a particular task. We give positive scores for meaningful actions such as joining, training, and evaluating FL tasks. Meanwhile, we give negative values to punish malicious behaviors. $k = 1, 2, 3, \ldots, K$ is the index of $C^{evt}$ where $K$ is the total number of distinct $C^{evt}$ types that is available in the system. $P$ is a threshold number to control how many cumulative credit events $C^{cum}$ can be stored for each client. The $t^{expired}$ is a threshold to justify the freshness of $C^{cum}$. We measure $t^{expired}$ in a block timestamp format.

The admin calls SAVECREDITEVENT($\cdot$) procedure to insert a new score for client $i$. This method should be called at the end of the FL task. First, we calculate $C^{cum}$ for a given client $i$. The $C^{cum}$ is based on the sum of all $C^{evt}$ that the client receives during the FL task. The $t^{cum}$ indicates the block timestamp when the $C^{cum}$ is calculated and stored in the blockchain.

Any entity (e.g., model owner, client, or reviewer) can call CALCULATECREDITSCORE($\cdot$) to receive the credit score of a given client $i$. The system first gathers all last $P$ of $C^{cum}$ and determines its freshness by comparing to $t^{now} - t^{expired}$. The outdated cumulative credit events will be ignored. The total credit score calculation is then scaled with $\frac{t^{expired} - (t^{now} - t_p^{cum})}{t^{expired}}$. This way, the system puts more attention on the most recent events. They are more critical than obsolete ones; therefore, we give them higher weight.

---

**Algorithm 2** Credit score system for client $i$

---

1: **on** startup:
2:     admin initiates $C^{evt}$, where $C^{evt} = \{C_1^{evt}, C_2^{evt}, C_3^{evt}, \dots, C_k^{evt}, \dots, C_K^{evt}\}$
3:         $C_1^{evt} = C^{join} = +1$                       ▷ successfully join the FL task
4:         $C_2^{evt} = C^{train} = +3$                   ▷ successfully train the FL model
5:         $C_3^{evt} = C^{eval} = +3$                  ▷ successfully evaluate the FL model
6:         $C_4^{evt} = C^{punish} = -10$         ▷ punishment if perform malicious actions
7:     admin sets $P = 5$               ▷ the number of $C^{cum}$ object to store
8:     admin defines $t^{expired} = 86400$         ▷ epoch time till $C^{cum}$ expires

9: **procedure** SAVECREDITEVENT($i$)
10:     gather all $C^{evt}$ for client $i$
11:     calculate $c_k$, where $c_k$ is the number of occurence for $C_k^{evt}$
12:     $C^{cum} = \sum_{k=1}^{K} c_k \times C_k^{evt}$
13:     $t^{cum} = t^{now}$
14:     store $C^{cum}$ and $t^{cum}$                   ▷ can save up to $P$ times

15: **function** CALCULATECREDITSCORE($i$)
16:     gather all $C^{cum}$ for the last $P$ for client $i$
17:     **for** $1, 2, 3, \dots, p, \dots, P$ **do**
18:         **if** $t_p^{cum} < (t^{now} - t^{expired})$ **then**
19:             $C_p^{cum} = 0$                   ▷ ignore outdated $C^{cum}$ score
20:     $C = \sum_{p=1}^{P} C_p^{cum} \times \frac{t^{expired} - (t^{now} - t_p^{cum})}{t^{expired}}$
21:     **return** $C$

---

### 3.2. Federated Learning Stage

The FL tasks can be divided into seven stages: start, commit-train, reveal-train, commit-eval, reveal-eval, aggregate, and end stage. All parameters presented here are provided as bare minimum requirements. They can be customized further according to the actual FL use cases.

#### 3.2.1. Start Stage

*FL Task Creation*: The model owner must first create a training smart contract along with its training properties. The model owner sets up training information and uploads it to the IPFS network, including, but not limited to: the initial global model $M^{(0)}$ with its sample dataset for testing $d^{test}$; the task type description $\tau^{desc}$ (e.g., supervised, unsupervised, or reinforcement learning); a timestamp $t^{now}$; nonce $\eta$. Formally, this process can be defined as follows.

$$Y_1^{ipfs} = IPFS(M^{(0)} \parallel d^{test} \parallel \tau^{desc} \parallel t^{now} \parallel \eta) \tag{1}$$

The model owner then uploads the task metadata to the smart contract, including, but not limited to: the IPFS hash of the task $Y_1^{ipfs}$—this hash also becomes the task id $\tau^{id}$; the task target $\tau^{target}$ (e.g., achieving 90% accuracy); the base deposit value $\beta$; the total reward for this task $R$; the minimum clients' level $l^{min}$; minimum reputation $C^{min}$ to join the task. Finally, the owner also puts a registration timeout $t^{regis}$, which indicates when the registration will be closed.

*Client Assignment*: At any given time, FL clients may join the tasks they are interested in. They can download the FL task information from the corresponding IPFS and smart contract. Based on all that information, they can analyze their capabilities to determine whether they can perform the task and earn profits.

For all clients $i = 1, 2, 3, \dots, I$ with $I$ is the total number of the available clients, the client $i$ must submit their address $\alpha_i$ and public key $PK_i$ to register the FL task. Moreover, the clients also need pay deposits $D_i$ to the smart contract $\mathcal{S}$, which is $F_{i \to \mathcal{S}}(D_i)$. The

deposit is calculated as follows, where $l_i$ and $C_i$ are the current client's level and credit scores (obtainable from Algorithms 1 and 2).

$$D_i = \beta + \frac{l^{min}}{l_i} \times \beta + \frac{C^{min}}{C_i} \times \beta \qquad (2)$$

Each client will pay a different amount of deposit from one another depending on their trustworthiness. In general, clients with higher levels and more honest behaviors (i.e., having higher credit scores) will receive discounts on their deposits, while the lower level and dishonest clients will pay more deposits. The discounts or markups scale based on their gaps toward $l^{min}$ and $C^{min}$.

When nearing $t^{regis}$ timeout, the model owner has options to prolong the registration step. The owner can extend the timeout if he cannot find enough suitable clients; however, it cannot be prolonged forever because the deposit must be returned to clients. If the owner feels satisfied with the registered clients, they can move on to the next steps. Otherwise, they can also cancel the task, and all clients will receive their deposits back.

### 3.2.2. Commit-Train Stage

The model owner queries the registration results from the smart contract, which includes a list of addresses $\alpha_i$ and public keys $PK_i$ of all registered clients.

*Distribution of Initial Global Model*: The model owner $\mathcal{O}$ creates a random secret key $k$. They then sign the key with $SK_\mathcal{O}$, which is $Z_1 = SIGN_{SK_\mathcal{O}}(k)$, and encrypt the key with $PK_i$. In particular, for all $i$, the owner performs $X_{1,i} = PKE_{PK_i}(k)$. The owner then uploads the encrypted keys to the IPFS.

$$Y_2^{ipfs} = IPFS(Z_1 \parallel \bigcup_{i \in I} X_{1,i}) \qquad (3)$$

The model owner then uploads the $Y_2^{ipfs}$ to the smart contract, which eventually notifies all clients that the key is ready in the given IPFS address.

*Local Training*: All clients query $Z_1$ and $X_{1,i}$ from IPFS using $Y_2^{ipfs}$. They then decrypt the key with $SK_i$ and verifies that $\mathcal{O}$ is the signer of $Z_1$. Formally, clients calculate $k = PKD_{SK_i}(X_{1,i})$ and $VER_{PK_\mathcal{O}}(k, Z_1) = True$. After that, the clients can begin to train $M^{(0)}$ using their own private dataset $d_i^{train}$.

Once the training completes, the clients produce the updated local models $M_i'$ with their associated results $T_i^{score}$. Before submitting the results, the clients must encrypts the trained model using $X_{2,i} = E_k(M_i')$ and signs the model using $Z_{2,i} = SIGN_{PK_i}(M_i')$. They then upload the encrypted model along with its signature to IPFS and generate commitment hashes $Y_{1,i}^{commit}$, which can be described as follows.

$$Y_{3,i}^{ipfs} = IPFS(X_{2,i} \parallel Z_{2,i} \parallel t^{now} \parallel \eta) \qquad (4)$$
$$Y_{1,i}^{commit} = H(Y_{3,i}^{ipfs}) \qquad (5)$$

After that, clients submits $Y_{1,i}^{commit}$ and $T_i^{score}$ to the smart contract; however, clients keep $Y_{3,i}^{ipfs}$ secret for now.

This stage completes when one of the following cases happens. First, when all clients have submitted $Y_{1,i}^{commit}$ and $T_i^{score}$ to the smart contract. Second, a stage timeout is triggered, which is indicated by $t_{commit}^{train}$.

### 3.2.3. Reveal-Train Stage

Before a $t_{reveal}^{train}$ timeout, clients need to disclose their $Y_{3,i}^{ipfs}$ from Equation (4) to the smart contract. The smart contract will verify that the revealed IPFS hashes match the ones previously submitted in the previous stage. When the hash matches, the smart contract

records the $Y_{3,i}^{ipfs}$ in the storage and continues to the next stage. If the hash does not match, the smart contract will punish the sender by depleting their deposit.

$$\text{with } Y_{1,i}^{commit'} = H(Y_{3,i}^{ipfs}), \begin{cases} \text{continue, if } Y_{1,i}^{commit'} = Y_{1,i}^{commit} \\ \text{punish } i, \text{otherwise} \end{cases} \tag{6}$$

### 3.2.4. Commit-Eval Stage

*Local Model Evaluation*: During this stage, all clients must validate each other training results and thus, become reviewers for other clients. We define the reviewers as $j = 1, 2, 3, \ldots, J$, where $J$ is the total number of reviewers.

First of all, reviewers download other clients' model from IPFS using $Y_{3,i}^{ipfs}$. They then perform the necessary decryption to obtain the model and verify its signature. Formally, for all $i$, the reviewers obtain $M_i' = D_k(X_{2,i})$ and make sure that $VER_{PK_i}(M_i', Z_{2,i}) = True$. All invalid models are ignored in the system. The reviewers then evaluate the $M_i'$ using the test dataset $d^{test}$. Once completed, the reviewers produce the $T_{i,j}^{eval}$ for all $i$, which are the evaluation scores for client $i$'s model from reviewer $j$. Before submitting the result, reviewers generate commitment hashes $Y_{2,j}^{commit}$ as follows.

$$Y_{2,j}^{commit} = H(T_{i,j}^{eval} \parallel \eta) \tag{7}$$

After that, the reviewer submits $Y_{2,j}^{commit}$ to the smart contract but keeps the value of $T_{i,j}^{eval}$ secret for now.

This stage completes when one of the following cases happens. First, when all reviewers already submitted $Y_{2,j}^{commit}$ to the smart contract. Second, a timeout is triggered which is indicated by $t_{commit}^{eval}$.

### 3.2.5. Reveal-Eval Stage

Before a $t_{reveal}^{eval}$ timeout, reviewers need to disclose their $T_{i,j}^{eval}$ and $\eta$ to the smart contract. The smart contract validates whether the revealed evaluation scores match the ones previously submitted in the previous stage. The smart contract will punish the clients by depleting their deposits if they do not match. When the hash matches, the smart contract saves the $T_{i,j}^{eval}$ in the blockchain and continues to the next stage.

$$\text{with } Y_{2,j}^{commit'} = H(T_{i,j}^{eval} \parallel \eta), \begin{cases} \text{continue, if } Y_{2,j}^{commit'} = Y_{2,j}^{commit} \\ \text{punish } i, \text{otherwise} \end{cases} \tag{8}$$

### 3.2.6. Aggregate Stage

After the smart contract obtains all the evaluation scores from all clients, the system can begin the model aggregation stage.

*Calculating Contribution Scores*: The CALCULATEWORKERCONTRIBUTION(·) function in Algorithm 3 is used to calculate the training contributions of client $i$ with respect to its corresponding evaluation scores $T_{i,j}^{eval}$. We calculate the first quarter $Q_1$, the second quarter (the median) $Q_2$, and the third quarter $Q_3$ of all $T_{i,j}^{eval}$. We then check if the previously claimed training score $T_i^{score}$ resides outside the boundary of $(Q_1 - IQR)$ and $(Q_3 + IQR)$. We assume that 50% of the client is always honest and 50% of submitted evaluation scores can be trusted. If $T_i^{score}$ is out of range, we punish client $i$. Finally, we return the median of all evaluation scores as an accepted training score for $M_i'$.

The CALCULATEREVIEWERCONTRIBUTION(·) function is used to measure the evaluation contributions of reviewer $j$ towards the model of client $i$. Similar to the previous function, we first calculate $Q_1$, $Q_2$, and $Q_3$ of all $T_{i,j}^{eval}$. We then calculate $d_{i,j}$, which is the difference between the submitted eval scores $T_{i,j}^{eval}$ from each $j$ towards the median $Q_2$. Because we assume that 50% of the client is honest, we expect that 50% of their submitted

evaluation scores will be closer to the median. We then normalize $d_{i,j}$ and flip the score so that the higher $d'_{i,j}$ values now become the score closer to the median instead of the lower ones. Finally, we ignore $d'_{i,j}$ that is outside the boundary of $(Q_1 - IQR)$ and $(Q_3 + IQR)$, and give them the lowest value possible.

---

**Algorithm 3** Processing contributions from client $i$ and reviewer $j$

---

1: **function** CALCULATEWORKERCONTRIBUTION($i$)
2:     $\forall j$, gather all $T_{i,j}^{eval}$
3:     calculate $Q_1$, $Q_2$, and $Q_3$ from all $T_{i,j}^{eval}$
4:     calculate $IQR = Q_3 - Q_1$
5:     get previously claimed training score $T_i^{score}$
6:     **if** $T_i^{score} < (Q_1 - IQR)$ **or** $T_i^{score} > (Q_3 + IQR)$ **then**
7:         punish worker $i$                    ▷ client submitted fake training score
8:     **return** $Q_2$

9: **function** CALCULATEREVIEWERCONTRIBUTION($j$)
10:     **for all** $i$ **do**
11:         $\forall j$, gather all $T_{i,j}^{eval}$
12:         calculate $Q_1$, $Q_2$, and $Q_3$ from all $T_{i,j}^{eval}$
13:         calculate $IQR = Q_3 - Q_1$
14:         **if** $T_{i,j}^{eval} < (Q_1 - IQR)$ **then**
15:             $d_{i,j} = |(Q_1 - IQR) - Q_2|$
16:             punish reviewer $j$                ▷ client submitted fake evaluation score
17:         **else if** $T_{i,j}^{eval} > (Q_3 + IQR)$ **then**
18:             $d_{i,j} = |(Q_3 + IQR) - Q_2|$
19:             punish reviewer $j$                ▷ client submitted fake evaluation score
20:         **else**
21:             $d_{i,j} = |T_{i,j}^{eval} - Q_2|$
22:     normalize and flip, $d'_{i,j} = 1 - \frac{d_{i,j} - min(d_i)}{max(d_i) - min(d_i)}$
23:     **return** $d'_{i,j}$

---

*Aggregating the Global Model*: In vanilla FL [2], we perform aggregation as follows.

$$w_{t+1} \leftarrow \sum_{i=1}^{I} \frac{n_i}{n} \times w_{t+1}^i \tag{9}$$

$n_i$ is the total number of data owned by client $i$ while $n$ is the total dataset from all clients. Using this formula, the aggregation weight is calculated based on the dataset ownership.

Meanwhile, we slightly modify the formula to adjust the weight based on the models' accuracy. More specifically, for all $i$, the owner calculates the following.

$$x_i = \text{CALCULATEWORKERCONTRIBUTION}(i)$$
$$x'_i = \frac{x_i - min(x)}{max(x) - min(x)}$$
$$w_{t+1} \leftarrow \sum_{i=1}^{I} \frac{x'_i}{\sum_{i=1}^{I} x'_i} \times w_{t+1}^i \tag{10}$$

The model owner gathers all median scores from reviewers. The owner then perform normalization on the scores $x'_1$ and aggregates the global model based on each client accuracy, which is weighted as $\frac{x'_i}{\sum_{i=1}^{I} x'_i}$. For this matter, we prefer a more accurate $x'_i$ rather than the less accurate one.

*Distributing Reward*: The model owner $\mathcal{O}$ splits the total rewards for workers and reviewers as follows.

$$R^{train} = \phi \times R, R^{eval} = (1 - \phi) \times R \tag{11}$$

$\phi$ is a parameter determined by the owner, where $0 \leq \phi \geq 1$.

To distribute the rewards for all workers, for each $i$, the owner performs:

$$F_{\mathcal{O} \rightarrow i}\left(\frac{x_i'}{\sum_{i=1}^{I} x_i'} \times R^{train}\right) \tag{12}$$

Note that the reward is weighted as $\frac{x_i'}{\sum_{i=1}^{I} x_i'}$, similar to the aggregation rules. Hence, the client that submits models with more accurate results will be rewarded more than the less accurate ones.

To distribute the rewards for all reviewers, for each $j$, the owner performs:

$$d_{i,j}' = \text{CALCULATEREVIEWERCONTRIBUTION}(j)$$

$$z_j = \frac{1}{I} \sum_{i=1}^{I} d_{i,j}'$$

$$F_{\mathcal{O} \rightarrow j}\left(\frac{z_j}{\sum_{j=1}^{J} z_j} \times R^{eval}\right) \tag{13}$$

$z_j$ is the average distance of evaluation scores. The reward is then weighted as $\frac{z_j}{\sum_{j=1}^{J} z_j}$ in which reviewers that submitted scores closer to the median scores will be given more rewards.

*Updating Clients' Level and Credit Scores*: The model owners give experience values to all clients that contribute to the FL task by invoking the GAINEXPERIENCE($\cdot$) procedure in Algorithm 1. When adding experience, the owner may also eventually increase the clients' level when the number of experience required is met. Note that malicious clients (e.g., dropping out or submitting fake scores) will not receive any experience.

Moreover, the owner also gathers all of the credit events $C^{evt}$ for all joined clients, calculates the cumulative credit events $C^{cum}$, and then saves the score in the smart contract using SAVECREDITEVENT($\cdot$) procedure in Algorithm 2. Honest clients will receive positive scores, while malicious clients will receive negative scores.

The clients' updated levels and credit scores will determine the number of deposits that the clients need to pay when joining future tasks.

### 3.2.7. End Stage

When all the steps in the aggregation stage finish, the owner will receive the updated global model, and the clients receive compensation for their efforts in training or evaluating the local models. The task now moves to the last stage, where no one can modify this FL task state in the smart contract. This frozen state is used for auditing or as a reference for future tasks related to this task.

## 4. Experimental Results and Analysis

*Setup*: We built a docker container utilizing 2 cores of CPU and 2 GB of RAM to run Ganache [11] and our decentralized application (dapp). The smart contract was written in Solidity language and was deployed to the Ganache using Truffle JS [12]. Node JS was used as the programming language to implement our dapp.

### 4.1. Assessing Client Credibility

We first analyze our proposed client credibility metrics, including leveling, credit score, deposits, model training score, model evaluation score, and reward distribution. Note that the given initial values in this paper are merely used as examples. There are no

right and wrong values here so developers can tweak and calibrate them according to their desired results.

### 4.1.1. Leveling System Results

Figure 2 plots the growth rate of clients' level based on the given $\gamma$ value (c.f., Algorithm 1). With higher $\gamma$, the number of required experiences to level up a client increases. Hence, developers can tweak this value to determine how fast clients can level up in their system. For example, with $\gamma = 1$, clients need to complete at least 200 FL tasks to reach level 20. In contrast, they need to perform more than 1600 FL tasks to reach the same level in $\gamma = 8$ setting.

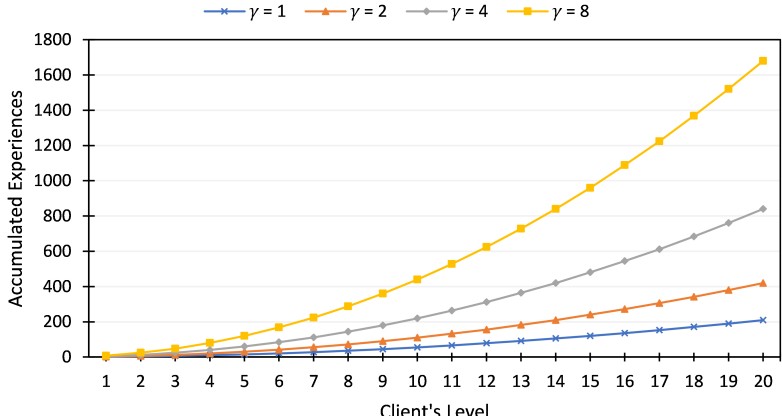

**Figure 2.** The accumulated experiences on each client's level depending on the $\gamma$ value.

### 4.1.2. Credit Score System Results

*Setup*: We make a simulated scenario where a client performs 20 FL tasks over a one-week timeframe. During that period, the client receives 20 score updates as presented in Table 2. Based on CALCULATECREDITSCORE($\cdot$) method in Algorithm 2, $P$ and $t^{expired}$ are the main variables to determine the freshness of the given credit events, and those value will influence the total credit score for a given client; therefore, we provide experiments with varying those values to see their impact on the total credit score. In the first two experiments, we set $t^{expired}$ to two days and then varied $P$ value to 2 and 10. In the last two experiments, we set $P$ to 5 and then varied the $t^{expired}$ to one day and three days. At each hour, we call the CALCULATECREDITSCORE($\cdot$) method and plot the results in Figures 3 and 4. The 20 straight vertical lines in both figures indicate when 20 score updates from Table 2 are applied in the system.

**Table 2.** The cumulative credit events added during one week timeframe using SAVECREDITEVENT($\cdot$) in Algorithm 1. These values are used to plot Figures 3 and 4.

| No | Credit ($C^{cum}$) | Date & Time ($t^{cum}$) | No | Credit ($C^{cum}$) | Date & Time ($t^{cum}$) |
|----|--------|-------------------------|----|--------|-------------------------|
| 1 | 7 | 23 April 2021 6:00 AM | 11 | 7 | 27 April 2021 2:00 AM |
| 2 | 7 | 23 April 2021 10:00 AM | 12 | 14 | 27 April 2021 5:00 AM |
| 3 | 14 | 23 April 2021 2:00 PM | 13 | −6 | 27 April 2021 9:00 AM |
| 4 | 14 | 23 April 2021 6:00 PM | 14 | −6 | 27 April 2021 5:00 PM |
| 5 | 7 | 24 April 2021 12:00 AM | 15 | −19 | 27 April 2021 7:00 PM |
| 6 | −6 | 24 April 2021 8:00 AM | 16 | 7 | 27 April 2021 10:00 PM |
| 7 | −6 | 24 April 2021 1:00 PM | 17 | 7 | 28 April 2021 6:00 AM |
| 8 | −19 | 24 April 2021 7:00 PM | 18 | 7 | 28 April 2021 9:00 PM |
| 9 | 7 | 26 April 2021 9:00 PM | 19 | 7 | 29 April 2021 10:00 AM |
| 10 | 7 | 26 April 2021 11:00 PM | 20 | 7 | 29 April 2021 4:00 PM |

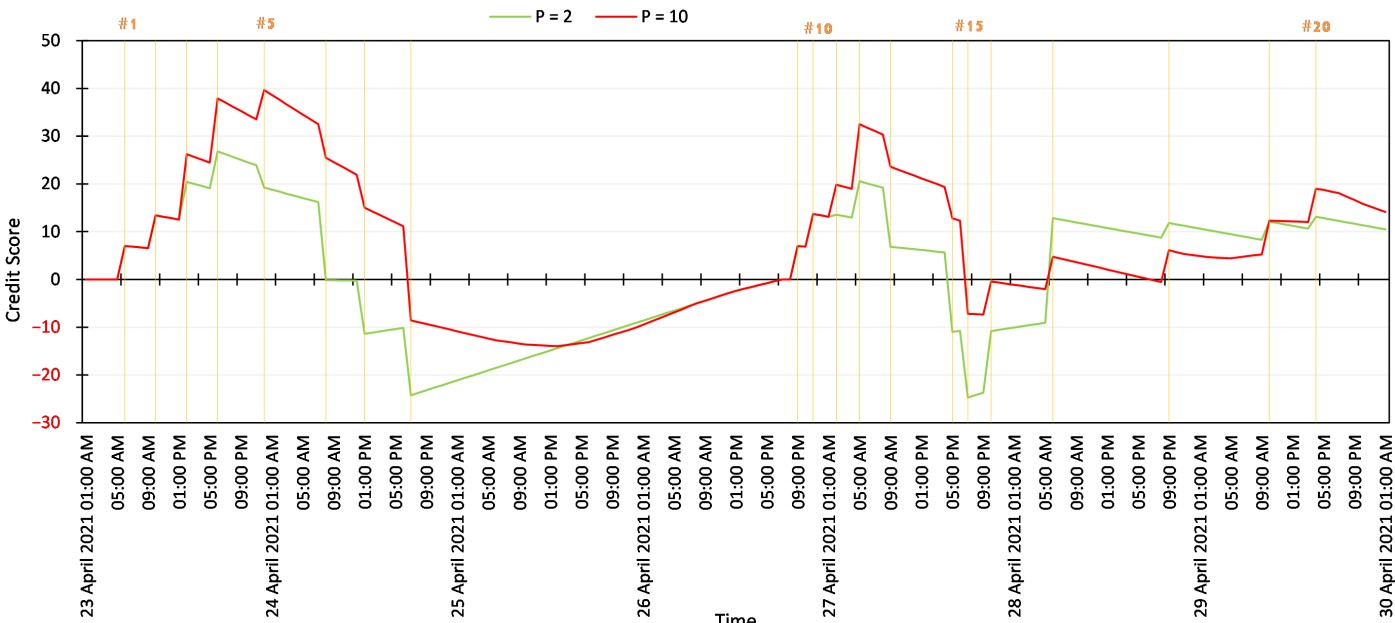

**Figure 3.** The fluctuating behavior of the client's credit score over one week timeframe when using $P = 2$ or $P = 10$ with $t^{expired}$ is set to two days

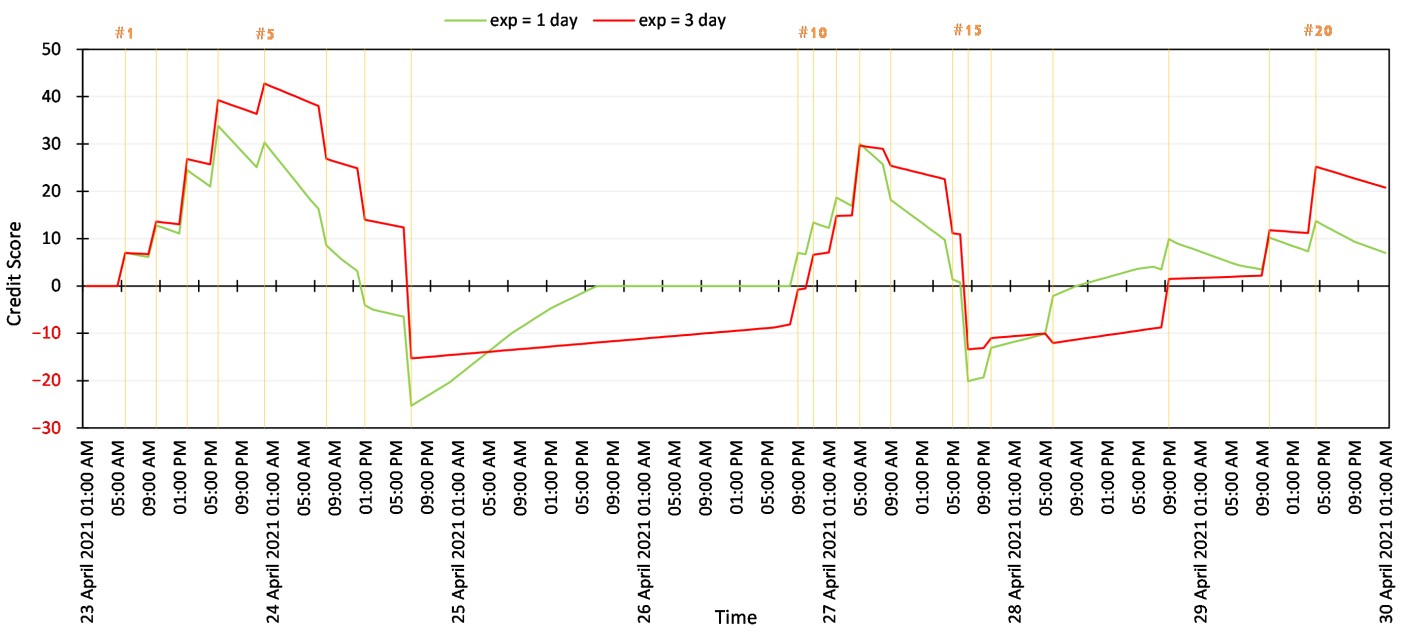

**Figure 4.** The fluctuating behavior of the client's credit score over one week timeframe when setting the expiry time to one or three days with $P = 5$.

When we set $P$ to a higher value, the system considers more events from the history to be included when calculating the current credit score. For $P = 10$, the system uses the last ten $C^{cum}$ to calculate the credit score, while only two $C^{cum}$ are used for $P = 2$; therefore, in Figure 3, we can see that the line for $P = 10$ becomes more positive than the $P = 2$ line from 23 April, 1 AM to 24 April, 1 AM. Similarly, we can see that the $P = 10$ line suffers less impact on the credit score drop than the $P = 2$ line from 24 April, 1 AM to 25 April, 1 AM.

When we set $t^{expired}$ to a higher value, the system will take a longer time for a credit score to return to its default value (zero score). We can see this behavior clearly in Figure 4 from 24 April, 6 PM to 26 April, 9 PM. In those periods, the $t^{expired} = 3$ day line takes a longer time to reach positive values than the $t^{expired} = 1$ day line.

Based on the results in Figures 3 and 4, we can conclude that our credit score can react to dynamics of client history activities by giving positives or negative scores. Moreover, we emphasize the freshness of the events, in which when clients become inactive, their reputation score will eventually return to zero.

Malicious clients with negative scores can improve their scores with two actions: (i) perform honest behavior to increase their score or (ii) do nothing and eventually let the credit score reset to zero. This behavior is intentional as we want to prevent clients from performing Sybil attacks by creating a new account with zero credit score value. Performing such action is not beneficial because adversaries will lose the previous client's level and start over again from level 1. The level is vital in our system as it will determine the amount of deposit the client needs to stake, which is explained in the following subsection.

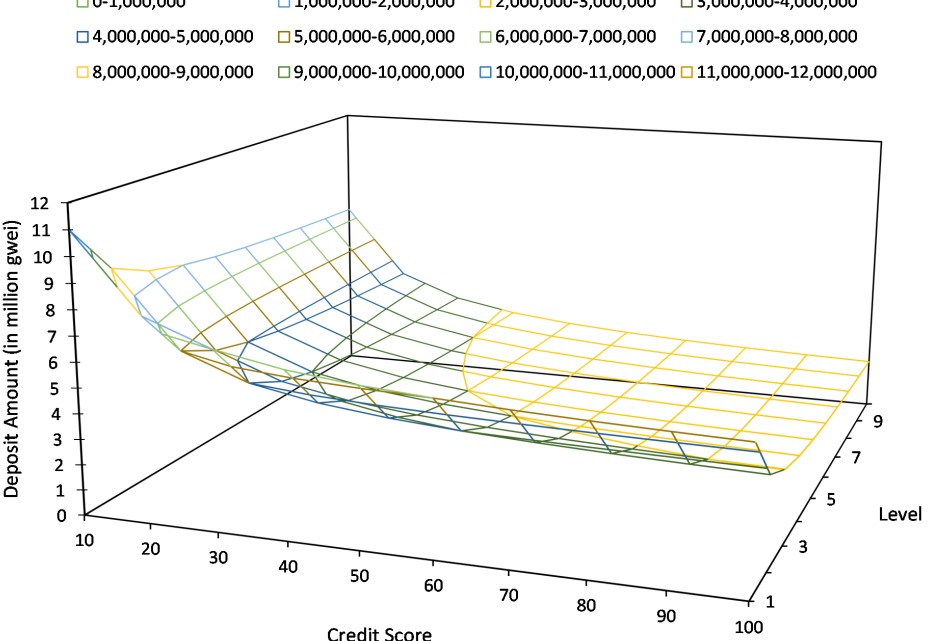

**Figure 5.** The amount of deposit (in million gwei) that clients need to pay depending on their current level and total credit score.

### 4.1.3. Deposit Mechanism Results

*Setup*: We first create a new FL task with the minimum level $l^{min}$ is set to 4, the minimum credit score $C^{min}$ is set to 60, and the base deposit $\beta$ is set to 1 million gwei. The number of deposits that the client has to stake depends on the client's current level and credit score; therefore, we simulate a scenario where multiple clients join the created FL task by varying the client level from 1 to 10 and the credit score from 10 to 100.

Figure 5 shows that the client with a lower level or credit score than the minimum requirements must pay a more considerable amount of deposits than the qualified and over-qualified ones. The increase can scale up to a triple amount of the normal deposit value, which should be closer to $3 \times \beta$ (c.f., Equation (2)). The yellow area (between 2 and 3 million) is a good deposit value.

### 4.1.4. Contribution Score Results

*Setup*: To simulate the effectiveness of our approach in determining workers' and reviewers' contributions, we deploy a simulation with ten clients. Client 3 is a malicious worker who sends fake training results. Furthermore, Client 3 and Client 6 act as malicious reviewers who submit fake evaluation scores to the system. We then run steps in Algorithm 3 and plot the results in Figures 6 and 7.

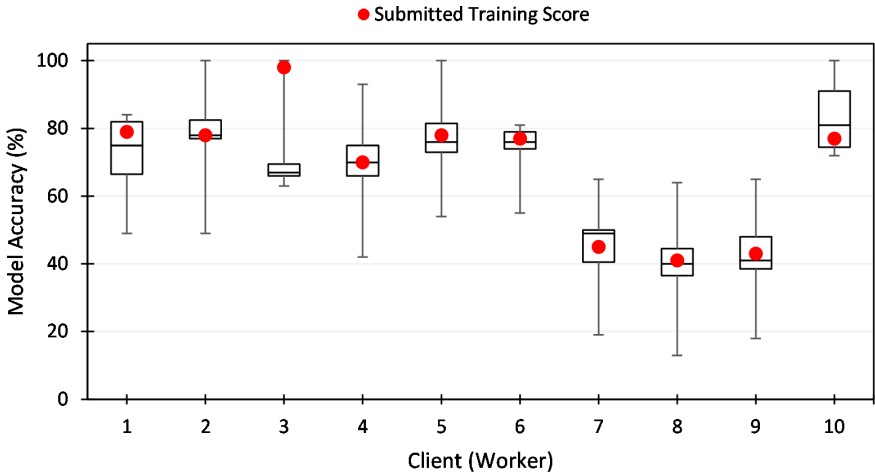

**Figure 6.** The distribution of model accuracy evaluation for each worker from the reviewers compared to the workers' claimed training score.

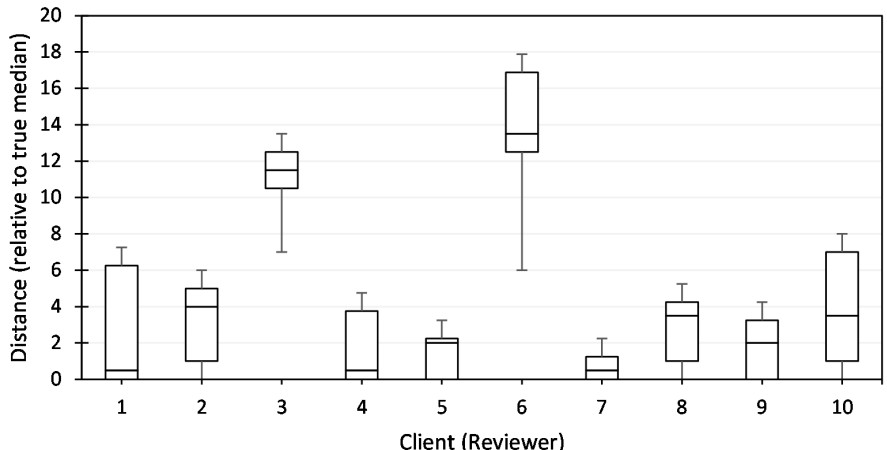

**Figure 7.** The distribution of submitted evaluation score from each reviewer compared to the distance toward the true median from CALCULATEWORKERCONTRIBUTION($\cdot$).

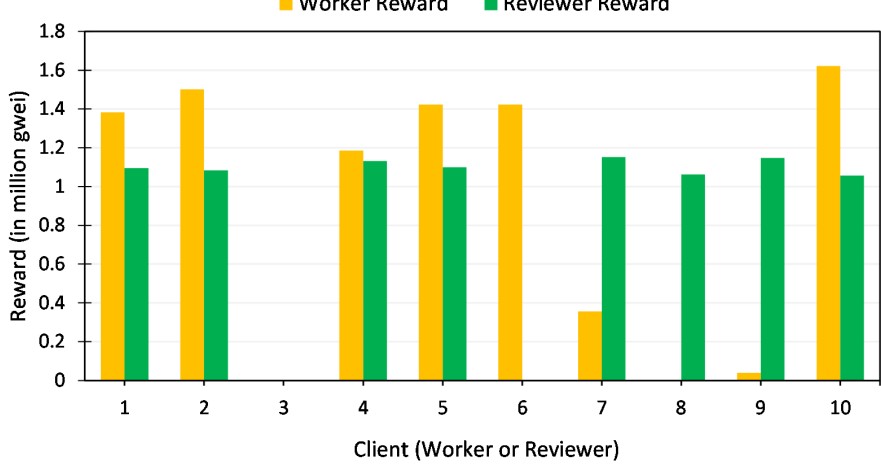

**Figure 8.** The amount of reward (in million gwei) that each trainer and reviewer receives depending on their contribution.

From Figure 6, we can see that the medians of each worker are placed relatively close to the submitted training scores previously claimed by the workers (except for Client 3). This condition remains true even though two malicious reviewers are in the system. Those

reviewers submit fake evaluation scores far from the ground truth, which explains the vast distance in each worker's min and max distribution in Figure 7. A greater distance means inaccurate evaluation and vice versa; therefore, as long as the majority of the reviewers (i.e., more than 50% clients) are honest, we can trust the result of our worker and reviewer contribution scores.

Moreover, from Figure 6, we can see that the claimed training score for Client 3 is far from the median as ground truth. This condition implies that Client 3 previously lied about the actual value of the training score and submitted a fake score in the system. Hence, the system can mark this client as a malicious worker. Furthermore, from Figure 7, the distance for Client 3 and 6 are higher compared to the rest. This fact indicates that those two clients submit evaluation results inaccurately. Thus, they are punished by the system.

Finally, based on the contribution results, we can distribute rewards to workers and reviewers as shown in Figure 8. In this case, we assume that the total reward for workers and reviewers is 20 million gwei, 10 million for workers, and another 10 million for reviewers. Clients 7, 8, and 9 receive low rewards because their models' accuracy is among the worst of all ten workers. Similarly, Client 3 does not receive any training rewards due to fake training result input, while Clients 3 and 6 do not receive any reviewer rewards due to their inaccurate evaluations.

### 4.2. Assessing Blockchain Performance

We analyze parts of our implementation that are related to blockchain.

#### 4.2.1. Gas Consumption

All Ethereum smart contract executions that modify the blockchain network state are subject to a unit called "gas". Generally, the more complex the methods become, the more gas is required to execute them. Table 3 shows gas consumption of our writable methods. Read functions are free and do not require gas; hence, we do not include them in the table.

**Table 3.** List of writable methods in our smart contract. We assume the block limit of 30 millions gas. $n$ indicates the total number of clients.

| Methods | Gas Usage per Tx | % Limit | # Tx per FL Task |
|---|---|---|---|
| Add Experience | 44,801 | 0.15 | $n$ |
| Save Credit Event | 102,944 | 0.34 | $n$ |
| Submit Training Score | 42,443 | 0.14 | $n$ |
| Submit Evaluation Score | 43,123 | 0.14 | $n(n-1)$ |
| Submit Commitment Hash | 48,234 | 0.16 | $n + n(n-1)$ |

First and foremost, all of our implemented methods are below the Ethereum gas limit standard of 30 million per block [13]. This indicates that all of our methods should be feasible to be executed in Ethereum networks. Second, those gas usages are calculated per one unit. For example, the add experience refers to adding new experience values to one client. Save credit event infer to submission of cumulative credit events for one client. Similarly, submission methods are for one submission of training score, evaluation score, and commitment hash. The total number of transactions for each method that must be executed in one FL task highly depends on the number of clients.

#### 4.2.2. Transaction Throughput

The transaction throughput in blockchain can be calculated as how many transactions the network can process per second (TPS). This metric depends on two factors: (i) number of transactions can be included in the block (which corresponds to the gas limit per block $g_{limit}$ in the Ethereum case), and (ii) the time taken to generate one block (also known as

block interval $b_{interval}$). With the results of the gas usage $g_{usage}$ per method in Table 3, we can estimate the projected TPS using the following formula

$$tps = (g_{limit}/g_{usage})/b_{interval} \qquad (14)$$

In this case, we assume that one block contains only transactions from the same methods.

Because block intervals vary among different blockchain networks, we consider three networks to measure the throughput: Mainnet (Ethereum main network using PoW [14]), Kovan Testnet (Ethereum test network using PoA [15]), and Klaytn (private Ethereum network using PBFT [16]). Mainnet processes one block every 13 s [17], Kovan Testnet can do it in 4 s [18], while Klaytn can form a block within 1 s [19].

As shown in Figure 9, the lower the block interval, the higher the transaction throughput becomes, and thus, we can complete more FL tasks. Those results show that our proposal depends heavily on the consensus algorithm that the blockchain employs, and they are most likely to become the main bottleneck in our system. In addition, the whole performance of our system also depends on how many clients join the FL task. From Table 3, we can see that, because of the requirements for clients to review each of the other trained models, they must upload evaluation scores and commitment hash multiple times for all reviewed clients. Thus, creating a massive bottleneck in our system, but the FL system becomes robust and fair.

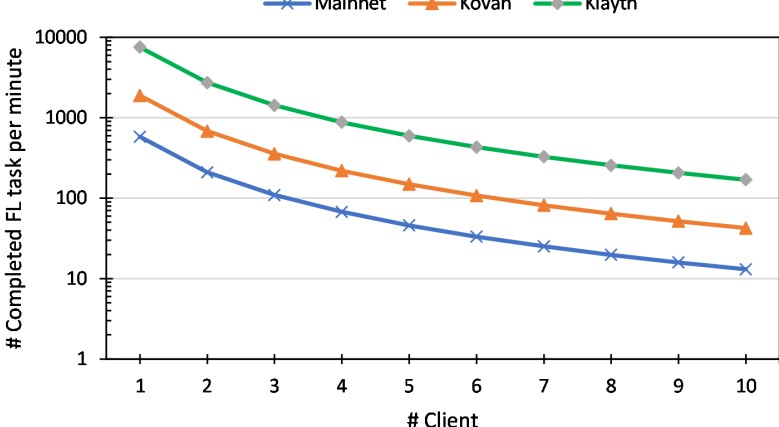

**Figure 9.** The total number of possible completed FL task per minute when executed in Mainnet, Kovan, and Klaytn network. This metric is calculated based on the block interval in Mainnet, Kovan, and Klaytn network for one round aggregation of local models.

*4.3. Assessing FL Accuracy*

Previous FL studies have conducted the accuracy of FL [2]. Moreover, previous blockchain-based FL system have analyzed that FL and blockchain run independently from each other [6,20,21]; therefore, the integration of blockchain to FL did not influence the accuracy result of FL. Because we do not propose a new FL algorithm in this paper, we refrain from discussing the FL performance in this paper.

## 5. Related Works

In recent years, many FL systems have been implemented in the Ethereum blockchain. Baffle [21] proposes a blockchain-based FL platform where workers can submit parts of the model to the smart contract for aggregation. The aggregation occurs on-chain; therefore, it does not need any centralized FL organizer. On the other hand, Morsbach [22] takes an off-chain aggregation approach by employing all workers to synchronize the global model through the IPFS network. Since storing the same model to the IPFS network will generate identical IPFS hashes, malicious entities can be detected, and most participants can quickly agree on the same global model. In [23], a blockchain-based FL system for healthcare is proposed. The system employs secure and private aggregation using multi-

party computation implemented in AMD Secure Encrypted Virtualization for maximum secrecy. While those studies have their own merits, they lack the deposit, incentive, or reputation system to control the workers' actions. Our proposal exists to provide all of the missing items.

The deposit system was used in several blockchain-based FL studies; however, most of them employ a static deposit system in which participants must submit a fixed amount of money to join the FL task, such as in [7,8,24]. CrowdSFL [6] proposes a dynamic deposit mechanism, where the required deposit is measured based on the age of the workers. A veteran worker may submit a lower deposit compared to a new worker. One limitation of this approach is that being veteran players does not guarantee they will not become malicious in the future; therefore, instead of relying on age alone, we also consider the worker's credit score to determine the deposit in our proposal.

Several studies have also explored the use of reviewers to judge trained local models. CrowdSFL [6] employs a Crowdsourcing Platform (CSP) to review the submitted local models from workers. The CSP evaluates the models and provides a guideline for the FL organizer on choosing the best model candidates for the model aggregation. Similarly, Kumar et al. [20] mandate all trained models to be reviewed. Models below a given threshold will be excluded from the aggregation process; however, these mentioned works employ a single reviewer approach, which suffers from centralization issues. In contrast, our proposal uses multiple reviewers.

When multiple reviewers exist, the workers disclose their models to all reviewers. The reviewers then evaluate the models using the test dataset and submit them to the smart contract. BlockFLA [24] proposes a penalty smart contract in which users can invoke it anytime they detect malicious training by providing proof of their malicious behaviors. The contract will determine which party is honest. Learning Markets [7] compares the difference between the claimed accuracy from workers and the ones from reviewers. The accuracy must surpass a given acceptance threshold to be included in the aggregation process and obtain rewards. BlockFlow [8] calculates the median of the evaluation scores from reviewers as a base truth to determine a valid accuracy value. This value is then compared with other scores to determine the quality of trained models. Though we share the same multi-reviewer architecture, the algorithm we use to calculate the contribution scores differs from those studies.

Finally, the reputation system is commonly used to track workers' actions throughout the workflow of FL tasks. CrowdSFL [6] and Learning Markets [7] have their reputation system built-in in their FL platform; however, their reputation calculation is simplistic as it only adds or subtracts the score with some fixed value. Our proposal proposes a more sophisticated credit score mechanism that also considers the data's freshness.

## 6. Conclusions and Future Works

This paper investigated several vanilla FL weaknesses that hindered its applications in untrusted environments and proposed counter solutions. We designed a blockchain-based FL aggregation protocol divided into seven stages and implemented the protocol in the Ethereum smart contract.

Through our evaluation results, we have shown that the leveling system can determine the longevity of a client, and the credit score evaluation could adapt to the history of client activities. Based on those two metrics, the proposed deposit mechanism could differentiate between trusted and untrusted clients by imposing different deposit requirements. Untrusted clients can pay deposits up to three times more than trusted clients. Furthermore, the proposed contribution assessment algorithm could detect malicious workers and reviewers to ensure the quality of training models. Proven malicious behavior will result in workers and reviewers losing 100% of their reward for a given FL task. Regarding scalability, we could complete up to 12 times more FL tasks in Klaytn than in Mainnet and about 2.25 times more in Kovan than in Mainnet; however, this performance would decrease exponentially as more clients join the FL tasks.

While our protocol is robust and fair, it creates a communication bottleneck in which clients need to send multiple transactions to the blockchain because of our peer-reviewed model. An improvement can be made such that instead of mandating all trained models to be reviewed, we can select to only review models from clients with low reputation scores; therefore, future works are required to find a good balance between robustness and performance. Furthermore, malicious clients can still collude outside of the blockchain. They may share their training or evaluation results off-chain, bypassing the commitment hash scheme we propose. Our system cannot defend against this attack because it is performed outside our system. For future work, we can combine the concept of homomorphic encryption with our proposal. All models become encrypted; thus, it becomes pointless for clients to share their trained models off-chain because they are encrypted with the system's homomorphic encryption public key and cannot be decrypted by clients.

**Author Contributions:** Conceptualization, Y.E.O. and B.S.; methodology, Y.E.O.; software, Y.E.O. and B.S.; validation, Y.E.O., B.S. and S.-G.L.; formal analysis, Y.E.O.; investigation, Y.E.O. and B.S.; resources, S.-G.L.; data curation, Y.E.O. and B.S.; writing—original draft preparation, Y.E.O.; writing—review and editing, Y.E.O., B.S. and S.-G.L.; visualization, Y.E.O.; supervision, S.-G.L.; project administration, S.-G.L.; funding acquisition, S.-G.L. All authors have read and agreed to the published version of the manuscript.

**Funding:** This work was supported by Basic Science Research Program through the National Research Foundation of Korea (NRF) funded by the Ministry of Education (Grant Number: 2018R1D1A1B07047601).

**Institutional Review Board Statement:** Not applicable.

**Informed Consent Statement:** Not applicable.

**Data Availability Statement:** Not applicable.

**Acknowledgments:** We want to thank the anonymous reviewers for their valuable comments.

**Conflicts of Interest:** The authors declare no conflict of interest.

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
