# Peer review of "Building Trusted Federated Learning on Blockchain"

_symmetry, doi:10.3390/sym14071407_

Round 1

Reviewer 1 Report

In this paper, the authors present a blockchain-based framework for the operation of federated learning. The proposed framework utilizes blockchain to provide a trusted environment that addresses several security issues that were not addressed in the federated learning implementations.

The paper generally holds merit. However, several issues need to be addressed before the paper is ready for publication.

Major Comments:
1. The paper is missing a "Discussion" section where the results can be compared to related works and critically evaluated. Once this section created, the analysis part of Section 4 can be moved to it.
2. The paper does not provide a "Related Works" section. Just mentioning references 6, 7, and 8 without any proper summarization and evaluation does not really help the reader formulate and idea about the importance of the work. There should be a proper section to discuss related works along with a summary table that compares the different features of these works with the proposed work.

3. I suggest moving the problem statement to Section 1 before the contributions. Then, sufficiently elaborate on the contributions rather than saying "solving the mentioned problems".

4. I also suggest moving the design considerations to the "Proposed Protocol" section because the proposed protocol design, and its design choices are highly relevant to this text.

Minor Comments:
1. Avoid using acronyms in the abstract.
2. The language of the paper needs proofreading. The paper readability is hindered in many places.
3. Consider looking for more relevant related works to include in the paper.

Author Response

We thank reviewers for their valuable feedback.

Please refer to the attached PDF file.

Author Response

(The authors gave the same response as above.)

Reviewer 3 Report

Just suggestions for revised  English language and

 check to spell

Author Response

(The authors gave the same response as above.)

Reviewer 4 Report

This paper runs federated learning using blackchain in untrusted domains. This paper is well organized and the contributions are fit for this journal. However, the following changes to be made before consider this paper for publication.

1. The literature of the paper is not found in the paper. It is recommended to provide the literature study using the recently published works and also highlight the limitations of them. Which of these limitations are addressed in the paper to be highlighted.

2. It is recommended to provide the contributions list in the Introduction. [2 of 19, line 43&44] are describing about the contributions, but they are not clear.

3. It is recommended to provide the computational complexities of the proposed algorithms (Algo.1-3) and Asymptotic complexity of the overall process also presented. It is also compared with the recent works.

4. Experimental results are well presented, however the authors must provide a discussion on reasons to get superior performance of the proposed work

5. Limitations of the proposed work to be studied in the paper.

Author Response

(The authors gave the same response as above.)

Round 2

Reviewer 1 Report

I would like to thank the authors for addressing most of my concerns.